# Prevalence and help-seeking for infertility in a population with a low fertility rate

**Mei-Chuan Lee[1,2], Pei-Shan Chien[1,3], Yue Zhou[1], Tsung Yu[1] ***

**1** Department of Public Health, College of Medicine, National Cheng Kung University, Tainan, Taiwan,
**2** Department of Pharmacy, Chi Mei Medical Center, Tainan, Taiwan, **3** Department of Chinese Medicine,
Chi Mei Medical Center, Tainan, Liouying, Taiwan

* tsungyu@mail.ncku.edu.tw

**Data Availability Statement:** Because our survey included sensitive questions, we cannot make our data publicly available. Data are available from the Institutional Review Board of the Chi Mei Medical Center, Taiwan (cmhirb@mail.chimei.org.tw) for

## Abstract

### Background

In Taiwan there has been limited research of epidemiological surveys on prevalence of infertility. This study aimed to provide the updated prevalence of primary infertility and of help-seeking among residents in Taiwan.

### Methods

Between February and March 2023, we conducted a cross-sectional population-based telephone survey of 1,297 men and women aged 20–49 years who were residing in Taiwan. We used computer-assisted telephone interviewing techniques to collect data regarding socio-demographic and reproductive characteristics. Using two approaches to defining infertility, we estimated the prevalence of infertility and the prevalence of help-seeking behaviors. Our analyses accounted for survey weighting.

### Results

The response rate was 27.9%. Among 1,297 respondents, 829 (63.9%) were married or cohabiting, including 404 men and 425 women. The prevalence of primary infertility using definition 1 was 5.6% (95% confidence interval [CI]: 4.2% - 7.4%); the prevalence of primary infertility using definition 2 was 6.7% (5.1% - 8.6%). Regarding professional help-seeking, 11.1% (9.2%-13.5%) had ever consulted a doctor about getting pregnant; 9.9% (8.1%-12.2%) had ever received diagnostic tests/treatment to help with conceiving; 2.6% (1.6% - 4.0%) were currently receiving diagnostic tests/treatment to help with conceiving.

### Conclusion

Our nationwide survey of the prevalence of primary infertility in Taiwan suggests that the prevalence was not as high as what is often seen in the news reports (about 14%). These findings also suggest there may be a gap between those who are currently experiencing infertility and those who are currently being treated; hence, we call for raising awareness of infertility and improving access to infertility healthcare.

researchers who meet the criteria for access to confidential data.

**Funding:** The study was funded by the National Science and Technology Council in Taiwan. The funders had no role in study design, data collection and analysis, decision to publish, or preparation of the manuscript.

**Competing interests:** None declared.

**Abbreviations:** CI, confidence interval; WHO: World Health Organization.

# Background

Infertility is defined as "a disease of the male or female reproductive system characterized by the failure to achieve a pregnancy after 12 months or more of regular unprotected sexual intercourse" [1]. Men and women or both may contribute to having difficulties in achieving a successful pregnancy. Infertility has many psychological consequences on the couples who are affected, including emotional distress, relationship strain, stigma, shame, and low self-esteem [2, 3]. Since infertility treatments and assisted reproductive technologies are very expensive, infertility also causes substantial financial burdens to many who are struggling [4]. On top of that, epidemiological studies have also pointed out the association of infertility with future disease risk such as cardiovascular diseases [5, 6] and cancers [7–9] for men and women alike.

In 2023, the World Health Organization (WHO) Department of Sexual and Reproductive Health and Research published a systematic review on the estimates of global and regional prevalence of infertility [1]. Their estimate of the lifetime prevalence of primary infertility (when a pregnancy has never been achieved) was 9.6% (95% confidence interval [CI]: 6.3, 14.3) and the period prevalence of primary infertility was 9.0% (95% [CI]: 6.6, 12.2). These estimates are important for learning the disease burden of infertility across countries and would be useful for establishing health policies regarding the prevention, diagnosis, and treatment of infertility. However, the WHO estimates of infertility varied greatly from country to country, and in some regions (e.g., Southeast Asia) no studies were even identified. The WHO report also did not identify any single study from Taiwan regarding the prevalence of infertility.

Taiwan is among the countries that are facing the challenges of rapidly declining fertility rates. In 2022, the total fertility rate, defined as the average number of children that would be born to a woman over her lifetime, was only 0.87 child per woman [10]. Recently, Chen and colleagues conducted a study to investigate factors associated with time to diagnosis of infertility in Taiwan and found that older age at marriage, higher educational level and higher gender equality scores were associated with earlier diagnosis while folk therapy use was associated with later diagnosis. Besides, the cost and affordability of infertility treatment are the major considerations when patients were making decisions [11]. Since 2021, the Ministry of Health and Welfare of the Taiwanese government has provided a subsidy for all married couples undergoing assisted reproductive technology treatments, with a view to increasing the access to infertility care [12]. Yet, there were not many epidemiological studies that estimated the prevalence of infertility of Taiwanese populations. To the best of our knowledge, only one article published decades ago and written in Chinese has addressed this research question [13].

We therefore conducted a population-based epidemiological survey in Taiwan to provide the updated estimates of the prevalence of primary infertility and the prevalence of couples' help-seeking behaviors.

# Materials and methods

## Data collection

We conducted a cross-sectional telephone survey of residents aged 20–49 years in Taiwan. The survey was administered from February 20th to March 2nd in 2023 by the Survey and Statistics Research Center at the National Cheng Kung University, Taiwan. In the beginning two days, we used random digit dialing to randomly sample landline telephone numbers in Taiwan. But later we switched to random digit dialing of mobile phone numbers in Taiwan since using mobile phone numbers was more efficient to collect our samples, and we had also better access to the younger age groups.

The principal investigator visited the Survey and Statistics Research Center to train the interviewers. When calling a phone number, the interviewer invited a person in the household who met the criteria (20–49 years of age) to participate in the survey. A total of 41,427 telephone/cellphone numbers were called; 4,648 were eligible for taking the survey. Among them, 1,297 respondents participated in our study: a 27.9% response rate (see **S1 Fig**).

The survey was anonymous and we obtained oral informed consent from all participants. Trained interviewers used Computer-Assisted Telephone Interviewing techniques to interview the participants for about 10 minutes and collected information on their sociodemographic and reproductive characteristics. After completing the survey, the participants were asked to provide their mobile phone numbers and we sent gift cards to them through text messages to thank them for their participation.

## Measures

Our survey questions were adapted from the National Survey of Family Growth in the United States and the Knowledge, Attitude, and Practice of Contraception Survey in Taiwan. All respondents were queried about their marital status and whether they were cohabiting with a partner. Among the 1,297 respondents, 829 (63.9%) were married or cohabiting. Among these, 219 (26.4%) had never had a child, but 13 of them reported that they or their partners were currently pregnant. We further classified the remaining 206 respondents as "infertile" or "childless and no pregnancy attempt".

We used two approaches to define infertility:

1. For the first definition (**Fig 1A**), respondents were considered infertile if they had been in a relationship longer than one year, and they reported regular sexual intercourse without any contraceptive methods in the past year, and they did not have a pregnancy in the past year. Those who had been in a relationship longer than one year and reported no regular sexual intercourse or were using any contraceptive methods in the past year were classified as "childless and no pregnancy attempt".

2. For the second definition (**Fig 1B**), respondents who had been in a relationship longer than one year and reported that they were still trying to become pregnant were classified as infertile. Those who reported not trying to become pregnant were considered "childless and no pregnancy attempt".

When we designed the questionnaire, we also asked respondents to report how long they had been trying to become pregnant, and we intended to use such information to perform the "current duration" analysis for the prevalence of infertility. However, most of these data that we collected seemed incorrect or were missing responses, and therefore we did not proceed with the current duration analysis [14].

To assess the help-seeking behaviors for infertility, respondents were asked whether they had ever consulted a doctor about getting pregnant and whether they had ever received or were currently receiving diagnostic tests/treatment to help with conceiving.

In our survey, educational levels were categorized into elementary, junior high, senior high, college, university, master's, and doctor levels. Family income levels were rated from 1 (very good) to 5 (very low).

## Statistical analysis

We conducted statistical analysis using STATA software (StataCorp LLC, College Station, Texas, USA) to account for the weighting of survey data (*svy* command). We obtained the

(A)

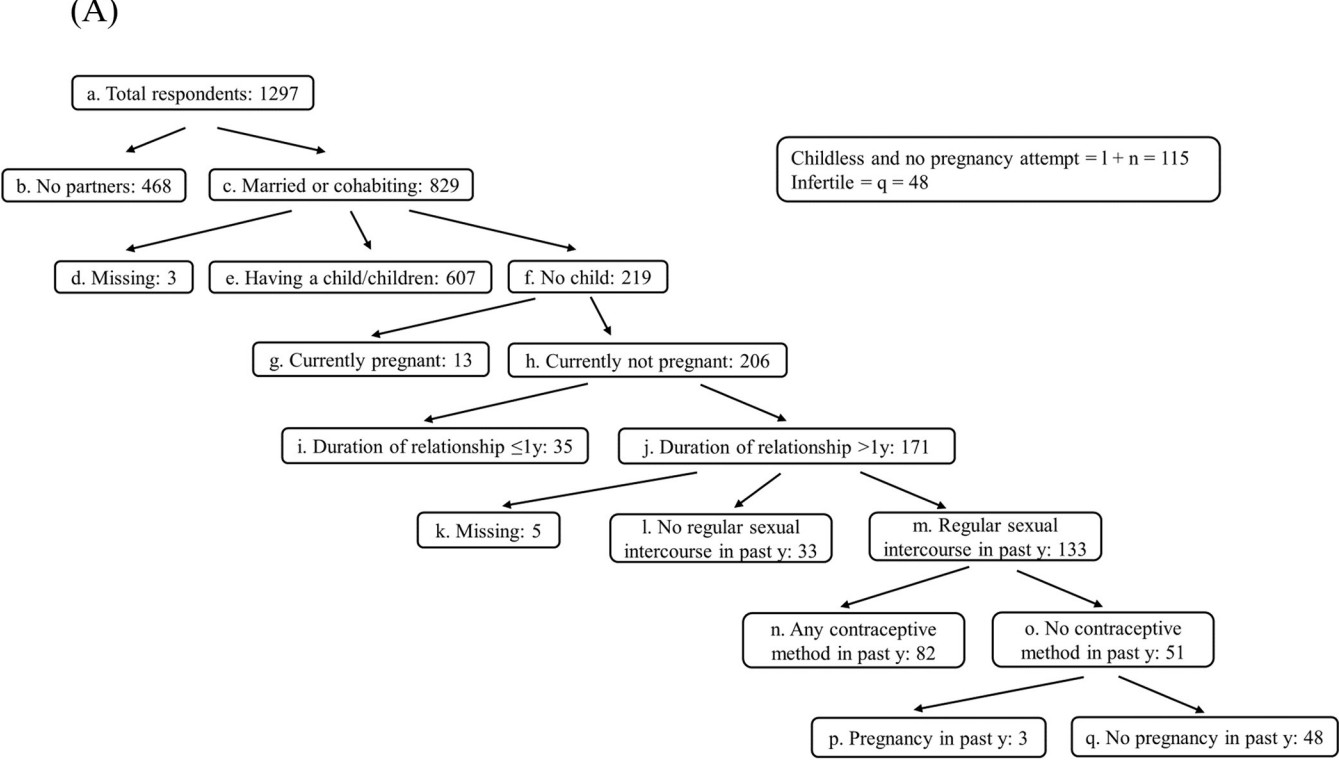

(B)

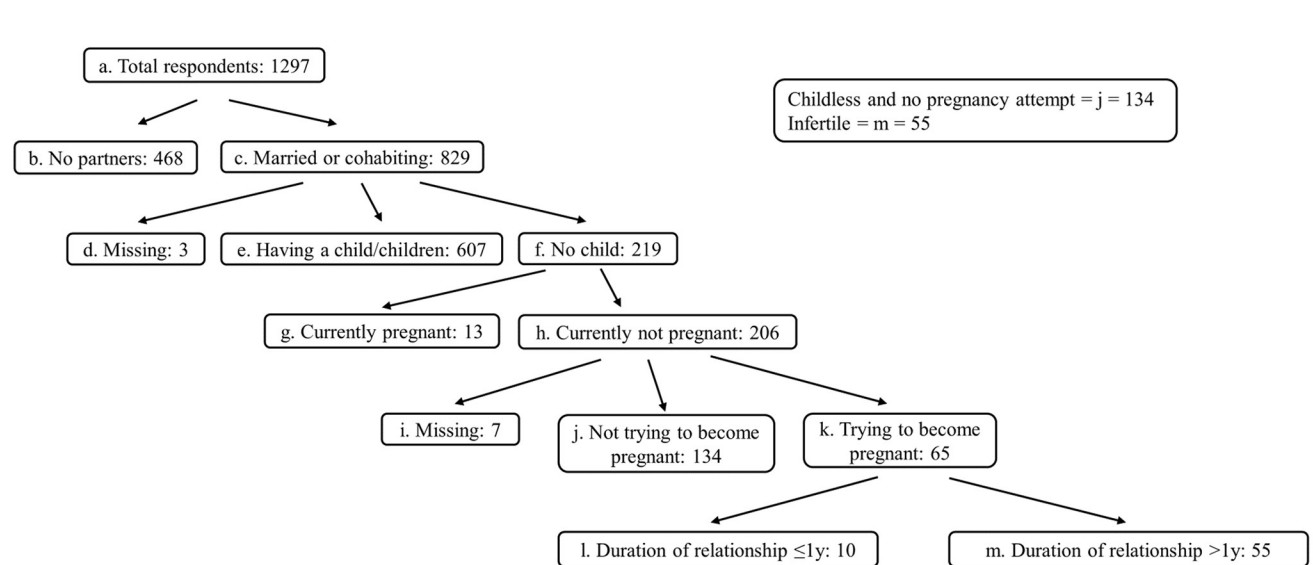

**Fig 1. Flow charts of the two approaches to defining respondents who were "infertile" and "childless and no pregnancy attempt".** **(A)** Definition 1. **(B)** Definition 2.

population distribution of Taiwan in March 2023 and post-stratified on age (20–24, 25–29, 30–34, 35–39, 40–44 and 45–49) and sex (men and women) groups to generate the survey weights. We estimated the prevalence of infertility and the prevalence of "childless and no pregnancy attempt" using the two definitions and the prevalence of help-seeking behaviors for infertility. The prevalence estimates were further stratified by age and sex groups.

### Ethical approval

The present study protocol was approved by the Institutional Review Board of the Chi Mei Medical Center, Taiwan (No. 11107–015).

## Results

Table 1 shows the characteristics of total respondents stratified by sex. There were 695 men and 602 women from 20 to 49 years of age. Most respondents had a university degree (weighted percentage: 51.6%). More women (56.4%) had a university degree than did men (46.8%). More men had a master's (15.4%) or doctoral degree (0.7%) than did women (11.2%

**Table 1. Characteristics of total respondents, stratified by sex.**

| Characteristics | Total respondents (n = 1297) | | Men (n = 695) | | Women (n = 602) | |
|---|---|---|---|---|---|---|
| | n | Weighted % (95% CI) | n | Weighted % (95% CI) | n | Weighted % (95% CI) |
| **Age (years)** | | | | | | |
| 20–24 | 85 | 13.3 (11.0–16.1) | 46 | 13.7 (10.5–17.7) | 39 | 12.9 (9.7–17.0) |
| 25–29 | 159 | 15.8 (13.7–18.2) | 88 | 16.2 (13.4–19.5) | 71 | 15.4 (12.4–19.0) |
| 30–34 | 235 | 15.9 (14.1–17.9) | 134 | 16.3 (13.9–19.1) | 101 | 15.5 (12.9–18.6) |
| 35–39 | 230 | 16.5 (14.6–18.6) | 124 | 16.5 (13.9–19.3) | 106 | 16.6 (13.9–19.8) |
| 40–44 | 302 | 20.1 (18.1–22.2) | 157 | 19.6 (16.9–22.6) | 145 | 20.6 (17.7–23.8) |
| 45–49 | 286 | 18.3 (16.4–20.4) | 146 | 17.7 (15.2–20.6) | 140 | 18.9 (16.2–22.0) |
| **Educational level** | | | | | | |
| Elementary | 6 | 0.4 (0.2–0.9) | 5 | 0.7 (0.3–1.6) | 1 | 0.1 (0.0–1.0) |
| Junior high | 26 | 1.7 (1.2–2.5) | 20 | 2.6 (1.7–4.0) | 6 | 0.8 (0.4–1.8) |
| Senior high | 256 | 19.9 (17.7–22.3) | 154 | 22.7 (19.5–26.2) | 102 | 17.1 (14.1–20.5) |
| College | 175 | 12.6 (10.9–14.6) | 81 | 11.1 (8.9–13.7) | 94 | 14.2 (11.7–17.2) |
| University | 643 | 51.6 (48.7–54.4) | 316 | 46.8 (43.0–50.7) | 327 | 56.4 (52.2–60.5) |
| Master | 184 | 13.3 (11.6–15.3) | 113 | 15.4 (12.9–18.3) | 71 | 11.2 (8.9–14.0) |
| Doctor | 7 | 0.4 (0.2–0.9) | 6 | 0.7 (0.3–1.6) | 1 | 0.1 (0.0–1.0) |
| **Marital status** | | | | | | |
| Single | 557 | 48.7 (45.9–51.6) | 338 | 53.8 (50.0–57.7) | 219 | 43.5 (39.3–47.8) |
| Married | 689 | 47.8 (45.0–50.7) | 332 | 43.0 (39.2–46.8) | 357 | 52.8 (48.6–57.0) |
| Divorced | 45 | 3.0 (2.3–4.0) | 23 | 2.9 (1.9–4.4) | 22 | 3.1 (2.1–4.7) |
| Widowed | 6 | 0.4 (0.2–0.9) | 2 | 0.3 (0.1–1.0) | 4 | 0.5 (0.2–1.5) |
| **Family income** | | | | | | |
| 1 (Very good) | 48 | 4.0 (3.0–5.3) | 20 | 3.1 (2.0–5.0) | 28 | 4.9 (3.3–7.1) |
| 2 | 363 | 28.8 (26.3–31.5) | 181 | 26.4 (23.1–30.0) | 182 | 31.2 (27.5–35.3) |
| 3 | 775 | 58.2 (55.4–61.0) | 435 | 61.8 (57.9–65.5) | 340 | 54.6 (50.4–58.8) |
| 4 | 85 | 6.4 (5.1–7.9) | 43 | 5.7 (4.3–7.7) | 42 | 7.0 (5.2–9.5) |
| 5 (Very low) | 26 | 2.6 (1.7–3.9) | 16 | 3.0 (1.8–5.0) | 10 | 2.3 (1.2–4.3) |

CI = confidence interval.

**Table 2. Prevalence of infertility and help-seeking among respondents who were married or cohabiting (N = 829).**

|  | Total (n = 829) | | Men (n = 404) | | Women (n = 425) | |
|---|---|---|---|---|---|---|
|  | n | Weighted % (95% CI) | n | Weighted % (95% CI) | n | Weighted % (95% CI) |
| Infertility (definition 1) | 48 | 5.6 (4.2–7.4) | 30 | 7.3 (5.1–10.3) | 18 | 4.2 (2.6–6.6) |
| Infertility (definition 2) | 55 | 6.7 (5.1–8.6) | 37 | 9.7 (7.0–13.2) | 18 | 4.2 (2.6–6.6) |
| Ever consulted a doctor about getting pregnant | 97 | 11.1 (9.2–13.5) | 48 | 11.6 (8.8–15.2) | 49 | 10.7 (8.2–14.0) |
| Ever received diagnostic test/treatment to help with conceiving | 86 | 9.9 (8.1–12.2) | 43 | 10.4 (7.8–13.9) | 43 | 9.5 (7.1–12.7) |
| Currently receiving diagnostic test/treatment to help with conceiving | 21 | 2.6 (1.6–4.0) | 13 | 3.1 (1.8–5.2) | 8 | 2.1 (1.0–4.4) |

CI = confidence interval.

and 0.1%, respectively). More men reported being single (53.8%) than did women (43.5%) and more women reported being married (52.8%) than did men (43.0%). Women tended to rate their family incomes higher than men.

Table 2 summarizes the main findings of this survey, listing the prevalence of infertility and help-seeking among the 829 respondents, including 404 men and 425 women who were married or cohabiting. The prevalence of primary infertility using definition 1 was 5.6% (95% confidence interval [CI]: 4.2%-7.4%) for all respondents, 7.3% (5.1%-10.3%) for men, and 4.2% (2.6%-6.6%) for women, respectively. The prevalence of primary infertility using definition 2 was 6.7% (5.1%-8.6%) for all respondents, 9.7% (7.0%-13.2%) for men and 4.2% (2.6%-6.6%) for women, respectively. With regard to professional help-seeking, 11.6% (8.8%-15.2%) of men and 10.7% (8.2%-14.0%) of women had ever consulted a doctor about getting pregnant; 10.4% (7.8%-13.9%) of men and 9.5% (7.1%-12.7%) of women had ever received diagnostic tests/treatment to help with conceiving; 3.1% (1.8%-5.2%) of men and 2.1% (1.0%-4.4%) of women were currently receiving diagnostic tests/treatment to help with conceiving.

Using definition 1 (Fig 2A and 2B) and definition 2 (Fig 2C and 2D), Fig 2 shows the proportion of men and women who were "infertile" and "childless and no pregnancy attempt" by age group. The prevalence of infertility for men peaked at age 30–34 using definition 1 and at age 25–29 using definition 2. The prevalence of infertility for women peaked at age 30–34 using both definitions 1 and 2. For "infertile" and "childless and no pregnancy attempt" combined, the prevalence peaked at age 25–29 for men and at age 20–24 for women. Fig 3A and 3B show the proportion of men and women regarding having "ever consulted a doctor about getting pregnant" and having "ever received diagnostic tests/treatment to help with conceiving". The proportion of those who had "ever consulted a doctor about getting pregnant" and had "ever received diagnostic tests/treatment to help with conceiving" peaked at ages 35–39 for men and at ages 45–49 for women.

## Discussion

There is a paucity of population-based surveys that have estimated the prevalence of infertility in Taiwan. Although some newspaper articles reported that one in seven (about 14%) couples were infertile in Taiwan [15], we were unable to identify any published epidemiological study that supported such claims. The only published scientific study we found was conducted by Tsai and colleagues in 1997 [13]. They surveyed 6,939 women who got married in April, 1990, and their estimate of the prevalence of primary infertility (no pregnancy after one year of regular unprotected sexual intercourse) was 6.41%. For people 20–49 years of age in Taiwan, our estimates of primary infertility using the two different definitions were 5.6% and 6.7%,

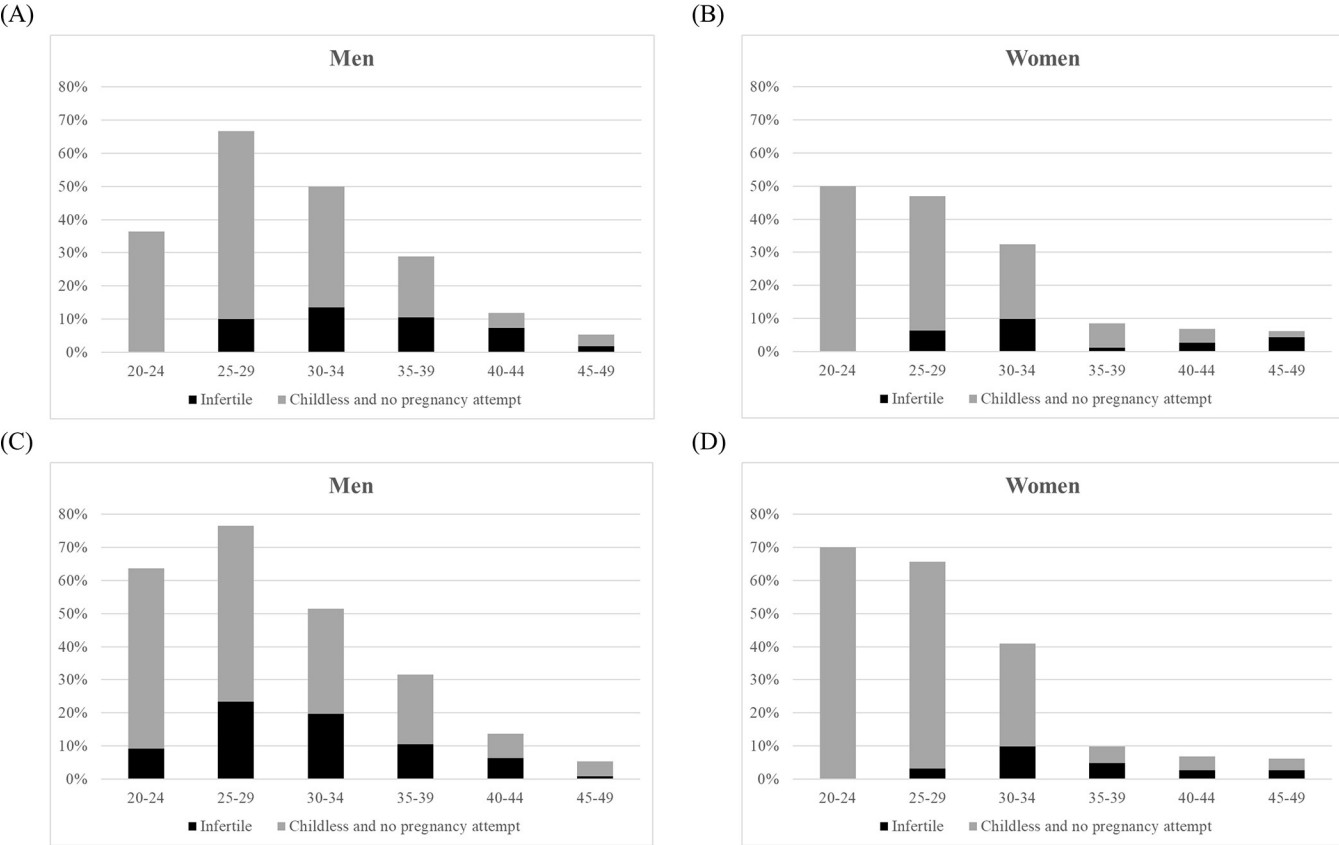

**Fig 2. Prevalence of respondents who were "infertile" and "childless and no pregnancy attempt," according to sex and age groups. (A)** and **(B)** were graphed using definition 1; **(C)** and **(D)** were graphed using definition 2.

respectively, which are close to the estimate reported by Tsai et al. That one in seven couples in Taiwan are infertile may possibly overestimate the actual prevalence of infertility.

Indeed, it is difficult to make comparisons between the prevalences of infertility reported by various studies conducted at different populations or time. The WHO report [1] on infertility prevalence systematically reviewed these estimates in the world and found that the

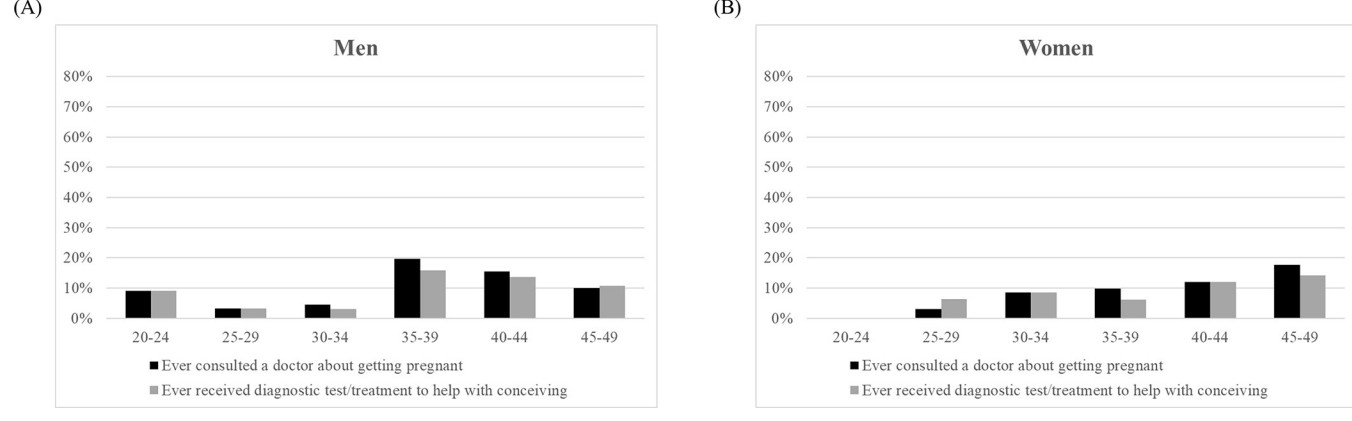

**Fig 3. Prevalence of help-seeking for infertility, according to sex and age groups. (A)** Men. **(B)** Women.

prevalence estimates were quite heterogeneous because of the populations studied and the methodology used. For example, the study populations may have included participants of different ages with different relationship status such as married, in union, or no restriction. The sample may be obtained from the general population or a clinic-based population. The definition of infertility may include a duration of pregnancy attempts ranging from 12 to 60 months. The methodology approach for estimating the prevalence of infertility may also vary, including prospective/retrospective time-to-pregnancy design [16, 17], current duration design [14], and self-reported/constructed infertility measures [18]. Moreover, a wide range of inclusion/exclusion criteria, survey instruments, and analytical approaches used by these studies make the comparison of infertility prevalence estimates even more challenging.

Our study estimated the "current period prevalence" of primary infertility that was the proportion of the population who had primary infertility at the time of the survey. The prevalence of "currently receiving diagnostic tests/treatment to help with conceiving" was 2.6%, which was lower than the prevalence of infertility (5.6% or 6.7%). This finding implies that some of the respondents with infertility problems might not have been aware of this condition or did not seek professional help. According to a population-based study in China [19], the main reasons for not seeking professional help may have included: (1) thinking that they have no fertility problems, (2) wanting to conceive naturally, and (3) having financial difficulties or are too busy for infertility treatments. It is thereby important to raise awareness of infertility among Taiwanese populations and guide those in need of professional medical help.

Our questionnaire did not intend to estimate the lifetime prevalence of primary infertility. Nonetheless, the prevalence of "ever consulted a doctor about getting pregnant" may be similar to the lifetime prevalence of total infertility, regardless of primary (when a pregnancy has never been achieved) or secondary (when at least one prior pregnancy has been achieved) infertility. The prevalence of "ever consulted a doctor about getting pregnant" was 11.1%; this estimate is likely to underestimate the true lifetime prevalence of infertility, since some of our respondents might not have sought professional help for their infertility.

A major strength of our study is that our survey sampled all residents from 20 to 49 years of age, including both men and women. Surveys of the prevalence of infertility that include men in the sample are uncommon; however, men also experience infertility as it is a couple-based condition and it has been suggested that men can reliably report information on a couple's infertility [20]. We were puzzled by our results that were stratified by age and sex (**Fig 3**). The prevalence of infertility was higher in men than in women, and by age group the prevalence of infertility peaked at around ages 30–34 years.

This could be explained by the following factors. Infertility is a condition that becomes a problem only when the couples are trying to conceive. In our study, men who were surveyed most likely had been married/cohabiting for a shorter period of time than their women counterparts of the same age, who might have gotten married at a much younger age. Therefore, more women than men of the same age might already have had children or were not trying to conceive anymore. It is also possible that women may have been less willing to disclose infertility problems than men due to the influence of stigma and culture [21]. Therefore, we need further studies to test for these hypotheses.

East Asian societies including Taiwan currently have the lowest fertility rate compared to any other part of the world. Is infertility to blame for the low fertility rate? We are not sure. Among the 1,297 respondents who were surveyed, not many (n = 829, 63.9%) were married or cohabiting. Among these 829 respondents, 115 (13.9%) or 134 (16.2%) individuals were classified as "childless and no pregnancy attempt," depending on the definition we used. We could not determine whether these people were infertile or not, given that they did not try to conceive. Increasing awareness of infertility problems and access to infertility care may be one

solution to increasing the fertility rate in Taiwan. However, people's declining willingness to form families and to raise children are the key drivers for the low fertility rate, as suggested by some sociological research [22].

Another strength of the present study is that we conducted a population-based telephone survey in Taiwan. Our response rate was not high (27.9%) but seemed reasonable because the previous systematic review on the prevalence of infertility found that more than half of the studies reported <75% response rate or failed to report a response rate [21]. If we had used medical records or health insurance data to estimate the prevalence of infertility, it would have underestimated the prevalence since people with infertility may avoid professional help and the national health insurance program in Taiwan does not cover all diagnostic tests/treatments for infertility. Besides, in a telephone interview as compared with a face-to-face interview, respondents may be more willing to answer sensitive questions regarding their sexual behaviors. Some of our survey questions, for instance, on "how long they had been trying to become pregnant," may not have been easy for our respondents to understand, interpret, recall, make judgement, and respond. Hence, we were unable to perform the "current duration" analysis due to the many errors we identified in their responses. This also reflects the difficulty in conducting a population-based survey of this kind.

In summary, our population-based survey of the prevalence of primary infertility in Taiwan suggests that the prevalence is not as high as what is often seen in the news reports. Our estimate of prevalence of infertility (5%-6%) was also lower than the estimate (9%) from the WHO report, but was close to the estimate in a previous epidemiological study in Taiwan. While conducting the survey, we encountered many challenges of estimating the prevalence of infertility and such limitations should be carefully considered when we interpret these results. These findings also suggest that there is a gap between those who are currently experiencing infertility and those who are currently being treated; hence, we recommend raising awareness of infertility and improving access to infertility healthcare.

## Supporting information

**S1 Fig. Recruitment and participation of telephone survey.**
(DOCX)

## Author Contributions

**Conceptualization:** Mei-Chuan Lee, Tsung Yu.

**Formal analysis:** Tsung Yu.

**Funding acquisition:** Mei-Chuan Lee, Tsung Yu.

**Investigation:** Pei-Shan Chien, Yue Zhou, Tsung Yu.

**Project administration:** Pei-Shan Chien, Yue Zhou, Tsung Yu.

**Supervision:** Pei-Shan Chien, Yue Zhou, Tsung Yu.

**Writing – original draft:** Mei-Chuan Lee, Tsung Yu.

**Writing – review & editing:** Mei-Chuan Lee, Tsung Yu.

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
