## [Decision Letter · Decision Letter 0]

5 Jun 2024

PONE-D-24-11968Prevalence and Help-Seeking for Infertility in a Population with a Low Fertility RatePLOS ONE

Dear Dr. Yu,

Thank you for submitting your manuscript to PLOS ONE. After careful consideration, we feel that it has merit but does not fully meet PLOS ONE’s publication criteria as it currently stands. Therefore, we invite you to submit a revised version of the manuscript that addresses the points raised during the review process.

We look forward to receiving your revised manuscript.

Kind regards,

Patrick Ifeanyi Okonta, MBBCh, MPH, FWACS, FMCOG, MD, DRH

Academic Editor

PLOS ONE

Journal Requirements:

"The study was funded by the National Science and Technology Council in Taiwan."

4. Thank you for stating the following in your Competing Interests section: "None declared."

5. In the online submission form, you indicated that "Please contact Prof. Tsung Yu for the study data and statistical codes."

**Additional Editor Comments:**

Author(s) should address the review comments of the three reviewers. In particular, authors should give 1. More details on the construction of the interview questionnaire 2. The reasons for the response rate and its probable effects on the validity of the results 3. Discussion and Conclusion based on the data from this study

Reviewers' comments:

Reviewer's Responses to Questions

**Comments to the Author**

1. Is the manuscript technically sound, and do the data support the conclusions?

Reviewer #1: No

Reviewer #2: Yes

Reviewer #3: Yes

2. Has the statistical analysis been performed appropriately and rigorously? 

Reviewer #1: No

Reviewer #2: Yes

Reviewer #3: Yes

3. Have the authors made all data underlying the findings in their manuscript fully available?

Reviewer #1: Yes

Reviewer #2: Yes

Reviewer #3: Yes

4. Is the manuscript presented in an intelligible fashion and written in standard English?

Reviewer #1: No

Reviewer #2: Yes

Reviewer #3: Yes

5. Review Comments to the Author

Reviewer #1: A. THE SOURCE OF THE SURVEY QUESTIONNAIRE IS NOT ATTACHED. IS IT A VALIDATED ONE OR NOT? -ALSO NOT MENTIONED!

B. (27.9% response rate is too small for a survey of this importance and to deduce any meaningful and representative conclusion or results.)

C. A response rate of 70% or more would have been more appropriate.

D. The conclusion is biased and inappropriate.

E. Infertility definitions 1 and 2; similar and very confusing. What is the relevance of this dichotomy in the definition? Could the authors elaborate on why they did not use only one definition for their analysis?

F. Outlandish discussion–not conclusively relevant to the analysis based on the data.

G. A need to correct all grammatical errors.

Reviewer #2: Good and relevant research. It was a cross sectional study done through telephone interviews which is adequate to get prevalence of a disease condition. The author however did not explore possible cultural and other sociological factors which may be the reason for the low fertility rate in Taiwan.

Reviewer #3: The manuscript is well written. technically sound and the statistical analysis is appropriate.

However can the conclusion be made to state whether the prevalence of infertility, based on the two operational definitions used for the study, was high or low in comparison to global, regional or national statistics. Anecdotal evidence from the press might not suffice.

6. PLOS authors have the option to publish the peer review history of their article (what does this mean?). If published, this will include your full peer review and any attached files.

Reviewer #1: No

Reviewer #2: **Yes: **HAJARATU UMAR SULAYMAN

Reviewer #3: No

---

## [Author Response · Author response to Decision Letter 0]

13 Jun 2024

Dear Prof. Okonta,

Thank you for reviewing our manuscript. The comments were very helpful for us to revise and improve the manuscript. Hereby please find below point-by-point responses to the reviewers’ comments. We also attached the revised manuscript with tracked changes and a clean version.

We look forward to seeing your further evaluation and response.

Sincerely yours,

Authors

Point-by-point response to the reviewers’ comments

Reviewer #1:

1. The source of the survey questionnaire is not attached. Is it a validated one or not? Also not mentioned.

Response:

Our survey questions were adapted from the National Survey of Family Growth in the United States and the Knowledge, Attitude, and Practice of Contraception Survey in Taiwan. We clarified this point in the Methods section (see lines 101-102 on page 8 in the manuscript with tracked changes).

2. 27.9% response rate is too small for a survey of this importance and to deduce any meaningful and representative conclusion or results. A response rate of 70% or more would have been more appropriate.

Response:

 We acknowledged the low response rate of our survey in the Discussion section (see lines 246-249 on page 15). It was indeed very difficult to get a high response rate in a telephone survey in Taiwan.

3. The conclusion is biased and inappropriate.

Response:

 We do not fully understand this comment.

4. Infertility definitions 1 and 2; similar and very confusing. What is the relevance of this dichotomy in the definition? Could the authors elaborate on why they did not use only one definition for their analysis?

Response:

 As what we explained in the Methods section (see lines 119-123 on page 9), when we designed the questionnaire, we also asked respondents to report how long they had been trying to become pregnant, and we intended to use such information to perform the “current duration” analysis for the prevalence of infertility. However, most of these data that we collected seemed incorrect or were missing responses, and therefore we did not proceed with the current duration analysis.

 As a result, we used the available data related to how long they had been trying to become pregnant to define infertility in another way, which was the definition 2 of our study.

5. Outlandish discussion–not conclusively relevant to the analysis based on the data.

Response:

 We do not fully understand this comment.

6. A need to correct all grammatical errors.

Response:

 We thank the reviewer for this comment. Accordingly, we asked for help of professional service to edit our manuscript.

Reviewer #2: 

7. Good and relevant research. It was a cross sectional study done through telephone interviews which is adequate to get prevalence of a disease condition. The author however did not explore possible cultural and other sociological factors which may be the reason for the low fertility rate in Taiwan.

Response:

 We thank the reviewer for this excellent suggestion. To explore the possible cultural and sociological factors that cause the low fertility rate in Taiwan was not the objective of the present study. In fact, we are also working on this issue in another manuscript.

8. Summary and Background:

The study was a cross sectional population-based study conducted through telephone interviews which sought to assess the prevalence of infertility as well as the number of people seeking help for fertility. This was a commendable study as it sought to determine the current state of fertility and fertility treatments in Taiwan given their low total fertility rate per woman which is less than 1. Though Taiwan was noted as being one of the countries facing a rapid decline in fertility, the study only concentrated on determining the prevalence of primary infertility; secondary infertility was not mentioned. The background of the study was silent about the cultural and religious beliefs regarding fertility in Taiwan as this would have given more insights the fertility seeking behavior of the people.

Response:

 Because of the questionnaire design we did not estimate the prevalence of secondary infertility, which was one of our study limitations. 

We provided a reference in the Background section relevant to the social determinants of infertility diagnosis and treatment in Taiwan (see lines 64-69 on page 5).

9. Methodology: Cross sectional studies are good in determining point prevalence and the number of people reached were above 40,000 which is commendable. However, the response rate of 27.9% was low and clear explanation for this was not given. In the first and second definitions of infertility used in this study, it was not clear if the use of contraception was the differentiating factor. Is it possible for ‘childless with no pregnancy attempt’ to have its own place in the definitions of terms in this study; it appeared in both groups?

Response:

 The response rate was not high although we tried our best to conduct the telephone survey. Please see our response to point 2.

 As for the definition of infertility, please see details in Figure 1. Contraception use was not the differentiating factor.

 “Childless with no pregnancy attempt” appeared in both definitions. We generated this group because this is a group of people who did not have children and were not trying to become pregnant. We were unable to know whether they were infertile or not.

10. Results and Discussion: With regard to professional help-seeking, 11.6% of men and 10.7% of women had ever consulted a doctor about getting pregnant; 10.4% of men and 9.5% of women had ever received diagnostic tests/treatment to help with conceiving; 3.1% of men and 2.1% of women were currently receiving diagnostic tests/treatment to help with conceiving’. In the above statement does it mean that male infertility is high in Taiwan? How does this compare to other Asian countries?

Response:

 Given that infertility is a “disease of couple”, our results do not imply that male infertility is higher than female infertility in Taiwan. Instead, our results suggested that the prevalence estimates of infertility obtained from male respondents were higher than female respondents in our study samples.

11. In the final recommendations in this study, the author wrote that: These findings also suggest that there is a gap between those who are currently experiencing infertility and those who are currently being treated; hence, we recommend raising awareness of infertility and improving access to infertility healthcare.’ I think the recommendation should go beyond this as there may be some other sociologic factors; be it cultural or based on the norm, which is hindering people from having children. This calls for a wider survey to get to the root cause why people who do not have children are reluctant to do so.

Response:

 We thank the reviewer for this excellent suggestion. As what we responded to point 7, to explore the possible cultural and sociological factors that cause the low fertility rate in Taiwan was not the objective of the present study. In fact, we are also working on this issue in another manuscript.

12. TO THE EDITOR: On a general note, this is a good study considering the paucity of studies from Taiwan and it can be considered for publication if the above issues are addressed. Thank you.

Response:

 We thank the reviewer for these positive comments.

Reviewer #3:

13. The manuscript is well written. Technically sound and the statistical analysis is appropriate. However can the conclusion be made to state whether the prevalence of infertility, based on the two operational definitions used for the study, was high or low in comparison to global, regional or national statistics. Anecdotal evidence from the press might not suffice.

Response:

 We thank the reviewer for this point. Accordingly, we added this point to the Discussion section (see lines 262-264 on page 16).

---

## [Editor Report · Decision Letter 1]

20 Jun 2024

Prevalence and Help-Seeking for Infertility in a Population with a Low Fertility Rate

PONE-D-24-11968R1

Dear Dr. Yu,

We’re pleased to inform you that your manuscript has been judged scientifically suitable for publication and will be formally accepted for publication once it meets all outstanding technical requirements.

Kind regards,

Patrick Ifeanyi Okonta, MBBCh, MPH, FWACS, FMCOG, MD, DRH

Academic Editor

PLOS ONE
---

## [Editor Report · Acceptance letter]

10 Jul 2024

PONE-D-24-11968R1 

PLOS ONE

Dear Dr. Yu, 

I'm pleased to inform you that your manuscript has been deemed suitable for publication in PLOS ONE. Congratulations! Your manuscript is now being handed over to our production team.

Kind regards, 

on behalf of

Professor Patrick Ifeanyi Okonta 

Academic Editor

PLOS ONE